# How School Support Influences the Content Creation of Pre-Service Teachers’ Instructional Design

**DOI:** 10.3390/bs15050568

**Published:** 2025-04-23

**Authors:** Yanlong Liang, Jijian Lu

**Affiliations:** 1School of Computer Science and Software, Zhaoqing University, Zhaoqing 526020, China; liangyanlong0918@163.com; 2Chinese Education Modernization Research Institute, Hangzhou Normal University, Hangzhou 311121, China

**Keywords:** school support, pre-service teachers, instructional design, generative AI technology, self-efficacy

## Abstract

In the rapidly evolving educational landscape, understanding how school support influences the content creation of pre-service teachers’ instructional design is crucial for fostering effective teaching practices and sustainable professional development. This study aims to explore the influence pathways and mechanisms through which school support affects the content creation of pre-service teachers’ instructional design. A total of 871 Chinese pre-service teachers were surveyed using an online questionnaire to assess school support, generative AI technology, self-efficacy, and instructional design content creation. The results indicate that school support has a significant positive predictive effect on the content creation of pre-service teachers’ instructional design. Moreover, generative AI technology and self-efficacy of pre-service teachers play a chain mediating role between school support and instructional design content creation. To enhance the content creation of pre-service teachers’ instructional design and promote the sustainability of teachers’ professional development, it is recommended that emphasis be placed on the application of school support and generative AI technology, as well as the enhancement of self-efficacy of pre-service teachers.

## 1. Introduction

Currently, generative artificial intelligence (AI) technologies, exemplified by ChatGPT, DeepSeek, Sora, ERNIE, and Xinghuo, which are based on large language models, have exerted a profound influence on human knowledge production, information acquisition, and educational paradigms. These technologies have gradually catalyzed an application revolution in the field of AI education ([56]). In contrast to traditional teaching environments, generative AI can effectively organize and summarize fragmented information scattered across the internet. It can also mimic human thought processes and language expression patterns, enabling it to process natural language dialogues and generate a variety of texts. This capability allows AI to assist teachers in managing routine and repetitive tasks, such as knowledge retrieval, multi-type text generation, and homework correction. By doing so, it helps to free up teachers’ time and energy, thereby stimulating their creativity. For instance, research has shown that the AI environment can influence the creativity of pre-service teachers in expanding instructional design scenarios ([32]). As the integration of intelligent technology and education deepens, the role of teachers must evolve. Pre-service teachers, in particular, need to shift their focus towards fostering students’ essential character traits, higher-order thinking skills, and complex problem-solving abilities. In this context, pre-service teachers are expected to become life mentors who guide students’ holistic development.

Instructional design is an activity that provides students with valuable tasks that enable them to achieve instructional goals and develop some of the skills that teachers want them to develop. In the teaching design, creating a relevant situation can expand students’ thinking and improve students’ ability to use knowledge. Creating a relevant situation is crucial for effective teaching. The innovation degree of the situation of creating in teaching design is an important standard to measure teachers’ creativity ([30]). It is important to accurately explore the influence of generative AI technology on the content creation of pre-service teachers’ instructional design and reveal its influence mechanism, promote the enhancement of pre-service teachers’ creative self-efficacy, and improve the efficiency of teaching work, so as to meet the development needs of more students, which not only conforms to the national education policy, but also helps to promote the digital transformation of higher education.

In this context, the rapid application of generative AI technology has attracted global attention. Generative AI technology is a technology that uses artificial intelligence algorithms to automatically generate new data or content. In the field of education, generative AI technology shows great potential. For example, through generative AI, personalized learning materials and exercises can be automatically created to meet the learning needs of different students, thus improving the teaching effect ([6]). In addition, the technology can also support teachers to develop interactive teaching resources ([38]). At present, in the research field of educators’ acceptance of large models, researchers mostly rely on models such as Theory of Planned Behavior (TPB) and Technology Acceptance Model (TAM) ([1]; [11]; [39]). As a result, factors such as perceived self-efficacy, perceived ease of use and perceived usefulness of AI by educators affect individuals’ willingness to use AI educational technologies ([22]; [15]). At the same time, a large number of studies show that school support is one of the important factors affecting teachers’ information technology integrated teaching ([7]). However, there are few studies on generative AI technology. Therefore, the three questions that this study attempts to address are the following:What impact does school support have on the content creation of pre-service teachers’ instructional design?Does the generative AI technology mediate the impact of school support on the content creation of pre-service teachers’ instructional design?Does the self-efficacy of pre-service teachers mediate the impact of school support on the content creation of pre-service teachers’ instructional design?Do the generative AI technology and pre-service teachers’ self-efficacy play a chain mediating role in the influence of school support on the content creation of pre-service teachers’ instructional design?

Therefore, this study collected the feedback of 871 pre-service teachers from Zhaoqing University in Guangdong Province by means of an online questionnaire. The independent variable was school support, the dependent variable was pre-service teachers’ teaching design content creation, and generative AI technology and pre-service teachers’ self-efficacy played a chain mediating role. Regression analysis was used to explore the influence of school support on the content creation of pre-service teachers’ instructional design, the application of generative AI technology, and the chain mediating role of pre-service teachers’ self-efficacy. This study examined the relationships among school support, generative AI technology, pre-service teachers’ self-efficacy, and pre-service teachers’ performance in content creativity of teaching design, so as to provide certain references for the application and implementation of intelligent teaching technology for pre-service teachers in China.

## 2. Literature Review

### 2.1. The Relationship Between School Support and Content Creation of Pre-Service Teachers’ Instructional Design

Teaching content design refers to a series of organizational activities designed with teaching objectives as the guide, centering on online teaching resources, knowledge content system, and teaching links before, during, and after class. Researchers mainly discuss the key factors affecting teachers’ instructional design content creation from the perspectives of individual traits and school environment. Relevant studies hold that factors such as teachers’ individual achievement motivation ([33]), self-efficacy ([26]), and satisfaction of self-psychological needs ([27]) are related to the exploration of teachers’ instructional content design. School-in-school support ([13]), innovative environment, and school resources are also important factors affecting the content creation of teachers’ instructional design. The support of schools is crucial to improving the teaching design ability of pre-service teachers. For example, the document “China’s Education Modernization 2035” ([36]) issued by the Ministry of Education emphasizes that systematic support measures, such as resource supply, technical support, and professional development, should be adopted to encourage teachers to use modern information technology to improve teaching methods, so as to effectively promote the growth of pre-service teachers in teaching design. The “Action Plan for Artificial Intelligence Innovation in Colleges and Universities” ([35]) document requires schools to be equipped with advanced hardware facilities, but also need to develop suitable teaching software and platforms. Through such support, pre-service teachers are able to explore in practice how technology can be integrated into curriculum design.

School support has a significant impact on teachers’ career satisfaction and teaching effect. Research shows that school organizational support enhances teachers’ job satisfaction and teaching effect by enhancing their enthusiasm to participate in educational research ([52]). Specifically, the school’s organizational support can be reflected in many aspects, such as providing teaching resources, building a culture of cooperation, and technical support. In short, teachers’ instructional design content creation is closely related to school support. It is concluded that school support plays a crucial role in improving teachers’ ability to create content in instructional design.

### 2.2. The Mediating Role of Generative AI Technology

With the development of technology, generative AI technology can provide a variety of support in education, including customized teaching content, providing powerful data analysis capabilities, and supporting personalized learning paths. The support of the school in terms of technical support and resource investment is crucial for the generative AI technology. Research has shown that technical training and support provided by schools can increase teachers’ confidence in the use of generative AI technology, leading to better implementation of these tools to improve the quality of education ([49]). A large number of relevant studies have shown that organizational support is an important factor affecting teachers’ work attitude ([14]; [41]; [21]; [16]), among which school support is an integral part of organizational support, so school support will affect teachers’ behavior in applying generative AI technology.

The study of [32] ([32]) further verified the role of generative AI technology and its application in promoting the content creation of instructional design. The research shows that the application of generative AI technology can not only meet the development needs of future teachers, but also provide them with targeted support environment. In addition, school organization and resource provision also play an important role in this process. Together, these factors create a creative and supportive environment, promote innovative activities, and ultimately improve the level of content creation of future teachers in instructional design ([54]). Therefore, generative AI technology has the potential to improve the level of teaching design content creation of pre-service teachers.

### 2.3. The Mediating Role of Self-Efficacy of Pre-Service Teachers

According to psychologist [3] ([3]), self-efficacy refers to an individual’s confidence in his or her ability to complete a certain task or a certain type of task. It is a self-perception of ability rather than a representation of actual ability. The combination of self-efficacy with specific practical activities of individuals produces new connotations, such as professional self-efficacy, teacher self-efficacy, etc. ([18]). In this study, teacher self-efficacy was defined as teachers’ belief in their ability to achieve desired results in student learning and engagement ([50]). Organizational psychology research shows that the support of an organizational environment has an impact on people’s emotions and behaviors ([19]). [2] ([2]) showed that schools with supportive cultures that encourage collaboration and professional development tend to foster higher levels of self-efficacy among teachers, and some studies have shown that positive work environments characterized by strong leadership and adequate resources contribute to higher self-efficacy among educators ([29]; [40]). In addition, in some studies, the leadership of school supervisors positively predicts teacher self-efficacy ([25]).

Positive psychological quality refers to the positive psychological characteristics that are formed on the basis of the interaction between individuals and the environment and are conducive to their own development ([31]). These psychological characteristics affect or determine the positive orientation of individual thoughts, emotions, and behaviors, thus improving individual behaviors. Therefore, self-efficacy has a great impact on the teaching innovation of pre-service teachers. Self-efficacy plays a crucial role in the teaching innovation of future teachers. Among teachers, a high level of self-efficacy can encourage them to persist in innovation, thus promoting the innovation of teaching design content creation ([4]). In this way, we expect that school support may enhance the content creation of pre-service teachers’ instructional design through their self-efficacy.

### 2.4. The Chain Mediating Effect of Generative AI Technology and Self-Efficacy of Pre-Service Teachers

As two key variables affecting the content creation of pre-service teachers’ instructional design, the generative AI technology and self-efficacy do not exist in isolation. In a large number of studies on generative AI technology, generative AI technology has an impact on teachers’ work attitude and performance ([42]). In terms of the relationship between the application of generative AI technology and self-efficacy, previous studies have found that there is a positive relationship between generative AI technology and teachers’ self-efficacy. Some studies have also found that generative AI technology also plays an important role in curriculum design. Through big data and algorithm analysis, teachers can design and adjust teaching content more efficiently, thus enhancing self-efficacy in curriculum design ([20]). It can be inferred that generative AI technology and self-efficacy of pre-service teachers are correlated with one another and that school support may indirectly affect the content creation of pre-service teachers’ instructional design through generative AI technology and self-efficacy of pre-service teachers.

## 3. The Research Hypotheses

In this study, school support may impact the content creation of pre-service teachers’ instructional design. However, the impact of school support on the content creation of pre-service teachers’ instructional design may not be solely direct, but could also be mediated by the generative AI technology and self-efficacy of pre-service teachers. Furthermore, generative AI technology may mediate the effect of self-efficacy of pre-service teachers on the content creation of pre-service teachers’ instructional design. Consequently, the mediating effect of school support on the content creation of pre-service teachers’ instructional design could be contingent upon the level the generative AI technology and self-efficacy of pre-service teachers.

In summary, the following hypotheses (see Figure 1) are proposed.

**Hypothesis 1 (H1).** 
*School group support has a positive predictive effect on the content creation of pre-service teachers’ instructional design.*


**Hypothesis 2 (H2).** 
*The application of generative AI technology mediates the relationship between school group and the content creation of pre-service teachers’ instructional design.*


**Hypothesis 3 (H3).** 
*Self-efficacy of pre-service teachers mediates the relationship between school group and the content creation of pre-service teachers’ instructional design.*


**Hypothesis 4 (H4).** 
*The application of generative AI technology and self-efficacy of pre-service teachers play a chain mediating role in the relationship between school group and the content creation of pre-service teachers’ instructional design.*


## 4. Methodology

### 4.1. Participants and Procedure

This study consists of two stages: pre-investigation and formal investigation. (1) Pre-investigation stage. Questionnaires were distributed online to pre-service teachers of relevant disciplines in Zhaoqing College. A total of 88 questionnaires were recovered, of which 78 were effective, with an effective rate of 88.6. SPSS 26.0 software was used to analyze the reliability of the pre-survey data. The Klonbach α coefficients of the overall scale and school support, the application of generative AI technology, the self-efficacy of pre-service teachers, and the content creation of pre-service teaching design were 0.923, 0.945, 0.803, 0.911, and 0.818, respectively. Among them, the factor load of individual items in the generative AI technology application table is <0.3, while the factor load of individual items in pre-service teachers’ instructional design content creation is <0.3. After deleting the factor load, the Klonbach α coefficients of generative AI technology application and pre-service teachers’ instructional design content creation are increased to 0.943 and 0.953. Therefore, individual items in the scale of application of generative AI technology and content creation of teaching design for pre-service teachers were deleted. In the exploratory factor analysis stage, the Harman single-factor test of common method bias in EFA showed that there were six factors with feature roots greater than 1, among which the maximum factor variance explanation rate was 24.637, which was lower than the critical standard of 40. Therefore, it was inferred that the probability of common method bias in pre-survey was small. Therefore, the modified questionnaire was used for formal investigation. (2) Formal investigation stage. The questionnaires were distributed in two ways: On the one hand, the questionnaires were distributed online to the pre-service teachers of relevant disciplines in Zhaoqing College, and a total of 549 questionnaires were collected; on the other hand, a total of 502 questionnaires were collected through relevant teachers distributing questionnaires in class. Finally, 180 invalid questionnaires were eliminated (invalid questionnaires were eliminated based on missing and empty questions; all the questionnaires were selected with the same option or regular), and 871 valid questionnaires were obtained, with an effective rate of 82.87%. The characteristics of valid samples are shown in Table 1.

### 4.2. Measurement

This study measured four groups of variables: school support, generative AI technology, pre-service teachers’ self-efficacy, and pre-service teachers’ instructional design content creation. All survey items were Likert-scale questions.

#### 4.2.1. School Support Scale

The social impact scale compiled by [48] ([48]) and revised by [51] ([51]) was adapted according to the use of generative AI technology by teachers and students in schools (see Appendix A). The revised scale includes four items, such as “I use generative AI technology because of the high proportion of teachers and classmates around the school who use generative AI technology”. All the questions were scored using a 5-level Likert scale. The higher the score of a student in a certain dimension, the stronger the sense of support in this dimension. The results of confirmatory factor analysis (χ2/df = 1.694, CFI = 0.999, NFI = 0.999, GFI = 0.998, TLI = 0.998, RMSEA = 0.028, SRMR = 0.004) showed that the factor load ranged from 0.799 to 0.921. The Klonbach alpha coefficient of internal consistency is 0.923, which indicates that the questionnaire has high reliability and validity.

#### 4.2.2. Self-Efficacy of Pre-Service Teachers Scale

For the measurement of self-efficacy of pre-service teachers, please refer to the Teacher Self-Efficacy Scale developed by [50] ([50]) and the General Teacher Self-Efficacy Scale designed by [45] ([45]). Based on the situation of pre-service teachers using educational AIGC to create instructional design content, this study has adapted five items (see Appendix A), such as “I am confident to do a good job in instructional design content creation.” All items were scored using a 5-level Likert scale, and the higher the score, the stronger the sense of self-efficacy. The results of confirmatory factor analysis (χ2/df = 1.518, CFI = 0.999, NFI = 0.998, GFI = 0.996, TLI = 0.999, RMSEA = 0.024, SRMR = 0.003) showed that the factor load ranged from 0.845 to 0.926. The Klonbach alpha coefficient of internal consistency is 0.955, which indicates that the questionnaire has high reliability and validity.

#### 4.2.3. Scale of Application of Generative AI Technology

The human-centered artificial intelligence application scale developed by [5] ([5]) and [23] ([23]) was adapted according to the application of generative AI technology in teaching design content creation. The revised scale contains four items (see Appendix A). All items were scored using a 5-level Likert scale. The higher the score, the stronger the willingness to apply generative AI technology. The results of confirmatory factor analysis (χ2/df = 1.686, CFI = 1, NFI = 0.999, GFI = 0.998, TLI = 0.999, RMSEA = 0.028, SRMR = 0.003) show that the factor load ranged from 0.859 to 0.918. The Klonbach alpha coefficient of internal consistency is 0.943, which indicates that the questionnaire has high reliability and validity.

#### 4.2.4. Instructional Design Content Creation Scale

The instructional design content creation questionnaire was compiled by [23] ([23]), and a total of 4 items were set to design the questionnaire (see Appendix A), all of which were scored using a 5-level Likert scale. The higher the score, the more conducive the application of generative AI technology was to the content creation of pre-service teaching design. The results of confirmatory factor analysis (χ2/df = 2.456, CFI = 0.999, NFI = 0.998, GFI = 0.997, TLI = 0.998, RMSEA = 0.041, SRMR = 0.003) show that the factor load ranged from 0.903 to 0.918. The Klonbach alpha coefficient of internal consistency is 0.953, which indicates that the questionnaire has high reliability and validity.

In this study, confirmatory factor analysis was used to evaluate the discriminative validity of the above variables. The results of confirmatory factor analysis, as shown in Table 2, showed a better fit (χ2 = 508.795, RMSEA = 0.063, NFI = 0.973, CFI = 0.979, GFI = 0.936) than the alternative measurement models (models 1 to 5). Therefore, the four-factor model can better represent the measured factor structure, indicating that school support, pre-service teachers’ instructional design content creation, generative AI technology, and pre-service teachers’ self-efficacy belong to four different constructs and have good discriminative validity.

Therefore, the benchmark model of this study is a four-factor model, which is specifically described as follows:

Model 1: School support + generative AI technology, and the three-factor model of pre-service teachers’ self-efficacy and the content creation of pre-service teachers’ instructional design;

Model 2: The generative AI technology + pre-service teachers’ self-efficacy, and the three-factor model of school support and the content creation of pre-service teachers’ instructional design;

Model 3: School support + pre-service teachers’ self-efficacy, and generative AI technology, the three-factor model of the content creation of pre-service teachers’ instructional design;

Model 4: School support + generative AI technology + pre-service teachers’ self-efficacy, and the two-factor model of the content creation of pre-service teachers’ instructional design;

Model 5: A single-factor model where all variables are combined into one factor.

### 4.3. Data Processing

In this study, SPSS 26.0 was used for quantitative analysis. First, a common method bias test was carried out to analyze whether there was common method bias in the data of this study. Then, descriptive statistics and correlation analysis were used to grasp the sample situation as a whole to reveal whether subjects with different sample characteristics had significant differences in major variables. Finally, the structural equation model was used to test whether the hypothesis proposed above is valid, and to investigate the interrelations among the independent variable of school support, the mediating variable of application of generative AI technology and pre-service teachers’ self-efficacy, and the dependent variable of the content creation of pre-service teachers’ instructional design.

## 5. Results

### 5.1. Common Method Deviation Test

Common methodological bias is likely to occur when single-source self-report questionnaires are used to collect data. According to the suggestions of [58] ([58]), the methods of “Harman single factor test” and “controlling unmeasured single method potential factor” were used to test the common method bias. The results of Harman’s single-factor test showed that the fitting effect of the single factor model (χ2/df = 14.146, RMR = 0.030, RMSEA = 0.123, CFI = 0.917, NFI = 0.911, TLI = 0.905, GFI = 0.763) was the worst. The fitting effect of the four-factor model was significantly better than that of the single-factor model, indicating that there was no serious common method bias. Then, after adding a method factor to the four-factor model, the fitting index of the model (χ2/df = 4.004, RMR = 0.011, RMSEA = 0.059, CFI = 0.985, NFI = 0.980, TLI = 0.978, χ2/df = 4.004, RMR = 0.011, RMSEA = 0.059, CFI = 0.985, NFI = 0.980, TLI = 0.978, GFI = 0.952) was better than the four-factor model, but χ2/df only increased by 0.497, and the improvement degree of other indices was between 0.003 and 0.006, and the fitting index did not improve significantly. This again showed that there was no significant common methodological bias among the variables measured in this study.

### 5.2. Descriptive Statistics and Correlation Analysis of Variables

The results of mean value, standard deviation, and correlation of the four main variables in this study are shown in Table 3. School support is significantly positively correlated with generative AI technology, pre-service teachers’ self-efficacy, and instructional design content creation. The content creation of instructional design is positively correlated with generative AI technology and the self-efficacy of pre-service teachers. Generative AI technology is significantly positively correlated with the self-efficacy of pre-service teachers. These correlations are consistent with the expectations of this study and provide preliminary support for subsequent hypothesis testing.

### 5.3. Multiple Stepwise Regression Analysis

In order to further explore the influence degree and effect of school support, generative AI technology, and pre-service teachers’ self-efficacy on instructional design content creation, a stepwise regression method was adopted to establish a linear regression equation with instructional design content creation as explained variable and other variables as explanatory variables. To examine the predictive effects of school support, generative AI technology, and pre-service teachers’ self-efficacy on the content creation of pre-service teachers’ instructional design, the results of regression analysis are shown in Table 4, and the corresponding regression equation is as follows:Teaching design content creation = 0.061 × school support + 0.644 × generative AI technology + 0.237 × pre-service teacher self-efficacy + 0.240

### 5.4. Direct Effect Test

From Table 5, with school support as the independent variable and pre-service teachers’ instructional design content creation as the dependent variable, the linear regression analysis shows that the direct effect size of school support on pre-service teachers’ instructional design content creation is 0.064 (*p* < 0.05), indicating that school support has a significant positive impact on pre-service teachers’ instructional design content creation. H1 is thus supported.

### 5.5. Test of the Mediating Effects

In this study, according to the results of correlation analysis, there was a significant correlation between school support, generative AI technology, pre-service teachers’ self-efficacy, and instructional design content creation, which supports further analysis. The mediation effect test method proposed by Hayes was adopted in this study. Using school support as independent variable, instructional design content creation as dependent variable, generative AI technology and pre-service teachers’ self-efficacy as mediating variable, the significance of mediating effect was tested. Model6, an Spss-based macro program PROCESS developed by him, uses the non-parametric percentage Bootstrap method with bias correction to extract 50,000 times repeatedly to estimate the 95% confidence interval. When the confidence interval of each path coefficient does not include 0, it indicates that the mediation effect is significant.

Table 5 shows the regression analysis of the relationship between variables in the chain mediation model. (1) In the school support → generative AI technology → pre-service teachers’ instructional design pathway, the positive predictive effects of school support on generative AI technology (*β* = 0.797, *p* < 0.001) and application of generative AI technology on pre-service teachers’ instructional design (*β* = 0.646, *p* < 0.001) are significant, thus suggesting that school support promotes pre-service teachers’ instructional design by generative AI technology. (2) In the school support → self-efficacy of pre-service teachers → pre-service teachers’ instructional design pathway, school support had a significant positive predictive effect on the self-efficacy of pre-service teachers (*β* = 0.343, *p* < 0.001), and the self-efficacy of pre-service teachers had a significant positive predictive effect on pre-service teachers’ instructional design (*β* = 0.239, *p* < 0.001), thus suggesting that school support promotes pre-service teachers’ instructional design by self-efficacy of pre-service teachers. (3) In the school support → generative AI technology → self-efficacy of pre-service teachers → pre-service teachers’ instructional design pathway, the positive predictive effect of generative AI technology on self-efficacy of pre-service teachers is significant (*β* = 0.636, *p* < 0.001), and the results indicate that the enhancement of generative AI technology facilitates the formation of self-efficacy of pre-service teachers. Generative AI technology and self-efficacy of pre-service teachers have significant chain mediating effects, and school support enhances the self-efficacy of pre-service teachers by increasing the application of generative AI technology, which in turn enhances pre-service teachers’ instructional design. In conclusion, H2~H3 are supported. In addition, the direct positive predictive effect of school support on pre-service teachers’ instructional design is significant, thus indicating that generative AI technology and self-efficacy of pre-service teachers partially mediate the relationship between school support and pre-service teachers’ instructional design (see Figure 2).

The pathway underlying the effect of school support on pre-service teachers’ instructional design is shown in Figure 1. The mediating effect was tested via bootstrap sampling, and the results revealed that the indirect effect via the path including application of generative AI technology as the mediating variable was 0.5048 (95% CI = [0.4336, 0.5762]), and the mediating effect accounted for 66.00%, assuming H2 is valid. The indirect effect via the path including the self-efficacy of pre-service teachers as the mediating variable was 0.0803 (95%CI = [0.0499, 0.1117]), and the mediating effect accounted for 10.50%, assuming H3 is valid. The indirect effect via the path including application of generative AI technology and self-efficacy of pre-service teachers as the mediating variables was 0.1187 (95%CI = [0.0694, 0.1729]), and the mediating effect accounted for 15.52%, assuming H4 is valid. Finally, the total of all indirect effects was 0.7038 (95% CI = [0.6355, 0.7678]), and the total mediating effect accounted for 92.02%. These findings suggest that the application of generative AI technology and the self-efficacy of pre-service teachers have both separate and serial mediating effects on the relationship between school support and pre-service teachers’ instructional design (see Table 6).

## 6. Discussion

On the basis of previous studies, this study proposed a model with school support as the predictor, generative AI technology and self-efficacy of pre-service teachers as the mediating variables, and the content creation of pre-service teachers’ instructional design as the result variables, showing a chain mediation effect. The main results show that school support directly improves the regulation of pre-service teachers’ instructional design content creation, and generative AI technology plays a mediating role in the association between school support and pre-service teachers’ instructional design content creation. Similarly, the self-efficacy of pre-service teachers also plays a mediating role in this relationship. In addition, the joint mediating effect of generative AI technology and pre-service teachers’ self-efficacy reveals the sequential mediating effect of school support on the regulation of pre-service teachers’ instructional design content creation. These findings provide a basis for further exploration and discussion. These findings provide a theoretical basis for future interventions aimed at improving content creation in instructional design through targeted psychological and technical support.

### 6.1. Direct Impact of School Support on the Content Creation of Pre-Service Teachers’ Instructional Design

The results show that there is a significant positive correlation between school support and content creation of pre-service teachers’ instructional design. In addition, the direct impact of school support on the content creation of pre-service teachers’ instructional design is substantial and provides strong support for H1. School support plays a multi-faceted role in the content creation of pre-service teachers’ instructional design, such as providing rich resources, systematic training, timely feedback, and practical opportunities, which can effectively promote the professional growth of pre-service teachers and the improvement in instructional design ability. This is consistent with the results of previous studies on school support; that is, the higher the level of school support, the higher the content creation level of instructional design ([44]).

In the rapidly evolving educational landscape, the instructional design capabilities of teachers are crucial for enhancing the quality of teaching. As technological advancements continue and societal expectations for educational excellence rise, teachers are compelled to continuously update their pedagogical approaches and content. Within this context, school support has emerged as a pivotal factor in facilitating the improvement in teachers’ instructional design content creation. Prior research has consistently demonstrated the significant role of school support in the professional development of pre-service teachers. For instance, studies have shown that technical support can substantially enhance teachers’ likelihood of utilizing digital tools in their instructional design ([17]). Similarly, other researchers have highlighted the critical importance of professional development opportunities for teachers to acquire and master new teaching methodologies ([47]). Furthermore, relevant investigations have underscored the impact of school culture on teachers’ innovative behaviors ([37]). Collectively, these findings reaffirm the essential role of school support in fostering teachers’ content creation within instructional design.

### 6.2. Mediating Role of Generative AI Technology

The standardized effect size of the indirect effect of path 1 “School support → Generative AI technology → The content creation of pre-service teachers’ instructional design” shows that generative AI technology significantly mediates the moderating effect of school support on pre-service teachers’ instructional design content creation. This finding suggests that generative AI technology is a key intermediary, and that school support enhances the application of generative AI technology, thereby improving the content creation supervision of pre-service teachers’ instructional design, confirming the effectiveness of H2.

Just as from the perspective of the technology acceptance model, the perceived usefulness and perceived ease of technology have a significant impact on the use of AI technology by schools and teachers in instructional design, and these factors will affect their behavioral intentions, and then affect the actual use behavior ([10]). This is consistent with the previous study conducted by [57] ([57]). In terms of content creation of instructional design, generative AI technology provides powerful auxiliary tools for pre-service teachers, who can use these tools to quickly create teaching materials that meet curriculum standards. Some studies have shown that school support plays a key role in promoting the practical application of generative AI technology in education ([9]). This study provides valuable evidence, which means that schools support providing an open and inclusive learning environment for pre-service teachers, encouraging them to try new technologies and apply generative AI technology. In such an environment, pre-service teachers are more confident in using generative AI technology to transform complex data into intuitive teaching insights, thereby creating richer, more vivid content in greater quantities.

School support includes comprehensive support including technology integration, AI education policy, teacher training and support, and resource allocation, which can effectively enhance the application of generative AI technology in education, and thus have a positive impact on instructional design content creation. School support requires teachers to actively encourage pre-service teachers to be innovative, show initiative, build self-efficacy, and apply generative AI technology in learning. Previous studies have indeed explored the correlation between the application of generative AI technology and the relationship between technology integration, education policy, teacher training and support ([53]; [24]), and the relationship between the application of generative AI technology and the content creation of pre-service teachers’ instructional design ([46]). However, there is a research gap on the mediating role of the application of generative AI technology in the relationship between school support and pre-service teachers’ instructional design content creation. Building on previous research, this study aims to contribute to the existing evidence by demonstrating that the application of generative AI technology is a mediating factor linking school support and content creation in pre-service teachers’ instructional design. These findings provide valuable insights into the internal processes that establish links between school support and content creation for pre-service teachers in instructional design.

### 6.3. Mediating Role of Self-Efficacy of Pre-Service Teachers

The standardized effect size of the indirect effect of path 2 “School support → Self-efficacy of pre-service teachers → The content creation of pre-service teachers’ instructional design” shows that school support significantly affects the content creation of pre-service teachers’ instructional design through pre-service teachers’ self-efficacy. This finding confirms that the self-efficacy of pre-service teachers plays a mediating role in the association between school support and content creation of pre-service teachers’ instructional design, supporting the validity of H3.

According to [12]’s ([12]) self-determination theory (SDT) ([34]), self-efficacy is a key factor affecting individual motivation and behavior. The findings also highlight the importance of cultivating pre-service teachers’ self-efficacy, as it is consistently associated with positive academic achievement and teaching activities. When schools provide pre-service teachers with ongoing professional development opportunities, positive feedback, and a supportive work environment, these teachers are more likely to develop higher self-efficacy. Teachers with high self-efficacy are more willing to try innovative teaching methods and technologies, and are more likely to devote more time and energy to designing high-quality teaching content ([28]). A study of pre-service teachers found that those who felt surrounded by school support reported higher levels of self-efficacy and they also performed better in instructional design ([8]), and these teachers tended to use more interactive and inquiry-based learning strategies, which not only captured students’ attention, but also facilitated the occurrence of deep learning. In addition, in the current education field, school support aspects such as technology integration, AI education policies, and teacher training and support are considered to be key factors to encourage the development of pre-service teachers’ self-efficacy and motivation ([43]), which are crucial to the design ability of pre-service teachers’ teaching activities. Therefore, this study builds on previous research to further understand how pre-service teachers’ self-efficacy moderates the relationship between school support and pre-service teachers’ instructional activity design. As such, these findings enhance the understanding of the internal processes that connect school support and content creation for pre-service teachers in instructional design.

### 6.4. The Chain Mediating Role of Generative AI Technology and Self-Efficacy of Pre-Service Teachers

According to the standardized effect size of indirect effect of path 3 “School support → Generative AI technology → Self-efficacy of pre-service teachers → The content creation of pre-service teachers’ instructional design”, school support significantly affects the instructional design content creation of pre-service teachers through the sequential intermediary of generative AI technology and self-efficacy of pre-service teachers. This finding validates H4, highlighting the complex role of generative AI techniques and self-efficacy of pre-service teachers in this relationship.

According to the perspective of self-determination theory, empowering teaching with the application of generative AI technology is an intrinsic demand of teachers ([55]). It is concluded that, the greater the school support, the higher the application of generative AI technology, the higher the self-efficacy of pre-service teachers, which will further affect the content creation of pre-service teachers’ instructional design. The application of generative AI technology in education can significantly improve the self-efficacy of pre-service teachers. Providing intelligent guidance and feedback through generative AI technology can help pre-service teachers plan and implement teaching activities more effectively, thus improving their teaching effectiveness and self-confidence. Self-efficacy plays a crucial role in influencing individual motivation and behavior, while the application of generative AI technology represents individual behavior. Specifically, self-efficacy can improve the application degree of generative AI technology by positively affecting individual motivation and behavior. To sum up, by encouraging pre-service teachers to be familiar with the application of generative AI technology and cultivating pre-service teachers’ self-efficacy, schools can indirectly affect the level of teaching design content creation of vocational education teachers. Therefore, schools should emphasize providing relevant support to pre-service teachers and encourage them to apply generative AI technology to improve their instructional design content creation throughout the teaching process.

## 7. Practical Implications

The present study extends the understanding of the multifaceted nature of content creation in pre-service teachers’ instructional design by identifying school support, the application of generative AI technology, and self-efficacy of pre-service teachers as key determinants. These findings hold significant implications for the training and education of pre-service teachers. From a practical standpoint, the results indicate that schools can enhance pre-service teachers’ instructional design content creation by fostering a supportive environment. Specifically, through technology integration, AI education policies, and targeted training programs, pre-service teachers can be effectively encouraged to take responsibility for instructional design content creation. This not only improves their instructional design capabilities but also ensures a more proficient use of technological tools in teaching, thereby enhancing teaching quality and student learning outcomes. Teachers should also focus on encouraging pre-service teachers to apply generative AI technology and cultivating pre-service teachers’ sense of self-efficacy, because these mediating factors play a crucial role in promoting the content creation of pre-service teachers’ instructional design. For researchers, this study highlights the importance of factors that influence the creation of instructional design content in pre-service teachers, and future research could explore the long-term effects of school support, pre-service teacher self-efficacy, and the application of generative AI technology on the creation of instructional design content and instructional activities in pre-service teachers. In conclusion, by recognizing the impact of school support, application of generative AI technology, and pre-service teacher self-efficacy on pre-service teacher instructional design content creation, educators and researchers can jointly implement targeted interventions to improve the overall quality of education. This research provides a solid foundation for further exploring the complexities of content creation in pre-service teachers’ instructional design and provides a framework for promoting its development in the classroom environment.

Building on the underlying assumption that school support plays a key role in facilitating the creation of instructional design content for pre-service teachers, this study makes several recommendations for strengthening this key practice area. First, educators should establish ongoing professional development programs to ensure that pre-service teachers are trained in the latest educational research and instructional techniques through regular workshops, seminars, and online courses to enhance their understanding of instructional design content creation and enhance their understanding and application of generative AI technology. Second, schools should consider the introduction of large-scale technical support and training, provide and promote educational technology tools, and regularly organize training to help pre-service teachers become familiar with and effectively use these tools, so as to enhance the interactive and effective teaching design. Third, schools design personalized training and guidance programs according to the level of self-efficacy of pre-service teachers. Pre-service teachers with high self-efficacy are more willing to adopt new technologies and teaching methods, while those with low self-efficacy may need more support and motivation to try new teaching strategies. Regular opportunities for reflection and feedback are provided in teaching practice courses to help pre-service teachers identify their strengths and weaknesses. It can make them more clearly understand the successful experience in teaching design and the need for improvement, so as to continuously improve their sense of self-efficacy.

## 8. Limitations and Future Study

Despite these encouraging findings, a number of deficiencies should be addressed to improve the validity and reliability of future investigations. First, the small sample size limits the external validity of the results, and future studies require a broader and more diverse group of participants. At the same time, the relevant factors of schools and local education authorities may have a certain moderating effect on the influence of pre-service teachers’ instructional design content creation. Looking forward to the future, we can further explore the independent and interactive effects of factors at the level of educational administrative departments and school factors on the content creation of pre-service teachers’ instructional design, and explore the internal mechanism of the relationship between school support and pre-service teachers’ instructional design content creation from a broader perspective.

## 9. Conclusions

The following conclusions were drawn under the conditions of this study: this study proposes a chain mediation model to explore school support and the content creation of pre-service teachers’ instructional design and its mechanism of action. The empirical research shows that, first, school support is significantly positively correlated with the content creation of pre-service teachers’ instructional design; that is, the higher the level of school support, the easier it is for pre-service teachers to achieve the content creation of instructional design. Second, the application of generative AI technology and self-efficacy of pre-service teachers have multiple (chain) mediating effects on the relationship between school support and the content creation of pre-service teachers’ instructional design via the following paths: the single mediating effect of the application of generative AI technology, the single mediating effect of self-efficacy of pre-service teachers, and the chain mediating effects of the application of generative AI technology and self-efficacy of pre-service teachers. That is, the higher the level of school support, the stronger the application of generative AI technology and the self-efficacy of pre-service teachers, the higher the level of design content creation of pre-service teachers; furthermore, school support promotes the application of generative AI technology, thereby impacting the level of self-efficacy of pre-service teachers and ultimately affecting the content creation of pre-service teachers’ instructional design.

## Figures and Tables

**Figure 1 behavsci-15-00568-f001:**
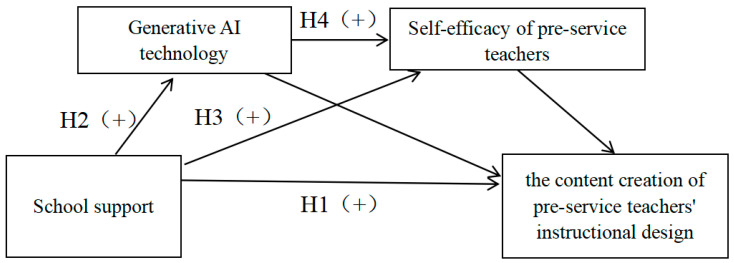
Research hypothesis model.

**Figure 2 behavsci-15-00568-f002:**
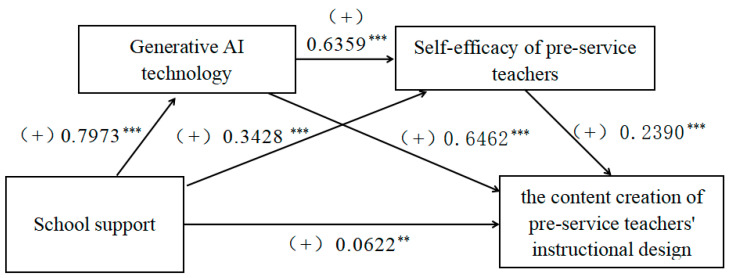
Chain mediation effect model. Note: ** *p* < 0.01, *** *p* < 0.001.

**Table 1 behavsci-15-00568-t001:** Demographic descriptive statistics of respondents.

Demographic Characteristics	Item	Frequency	Ratio
Sex	Male	383	42.97%
	Female	488	56.03%
Domicile	Rural area	654	75.09%
	City	217	24.91%
Grade	Sophomore	332	38.12%
	Junior	212	24.43%
	Senior	327	37.54%
Major	Science	233	26.75%
	Science and Technology	310	35.59%
	Mathematics	328	37.66%

**Table 2 behavsci-15-00568-t002:** Comparison of measurement models.

Models	χ2	df	χ2/df	CFI	TLI	NFI	GFI	RMESEA	SRMR	Δχ2
Benchmark Model (M1)	508.795	113	4.501	0.979	0.975	0.973	0.936	0.063	0.015	
Model 1	1211.683	116	10.446	0.942	0.931	0.936	0.828	0.104	0.027	702.888
Model 2	753.056	116	6.492	0.966	0.960	0.960	0.899	0.079	0.019	244.261
Model 3	978.128	116	8.432	0.954	0.946	0.948	0.865	0.092	0.023	469.333
Model 4	1322.780	118	11.210	0.936	0.926	0.930	0.816	0.108	0.027	813.985
Model 5	1683.395	119	14.146	0.917	0.905	0.911	0.763	0.123	0.030	1174.600

**Table 3 behavsci-15-00568-t003:** Descriptive statistics and correlations of variables.

Variables	M ± SD	1. School Support	2. Generative AI Technology	3. Self-Efficacy of Pre-Service Teachers
1. School support	3.06 ± 0.812			
2. Generative AI technology	3.22 ± 0.798	0.797 **		
3. Self-efficacy of pre-service teachers	3.17 ± 0.801	0.850 **	0.909 **	
4. The content creation of pre-service teachers’ instructional design	3.25 ± 0.795	0.781 **	0.913 **	0.879 **

Note: n = 871, ** *p* < 0.01, applicable to all statistical results in this article.

**Table 4 behavsci-15-00568-t004:** Results of regression analysis.

Variables	Non-Standard Regression Coefficient		Standardized Coefficient	*t*	*p*	*R* ^2^	Δ*R*^2^
	β	Standard Deviation					
School support	0.061	0.025	0.062	2.468	0.014	0.921	0.849
Generative AI technology	0.644	0.032	0.646	20.247	0.000		
Self-efficacy of pre-service teachers	0.237	0.036	0.239	6.539	0.000		
Constant	0.240	0.045		5.355	0.000		

**Table 5 behavsci-15-00568-t005:** Regression analysis of the relationship between variables in the chain mediation model.

Regression Equation		Overall Fitting Index			Significance of Regression Coefficient	
Outcome Variable	Predictive Variables	*R*	*R* ^2^	*F*	*β*	*t*
Generative AI technology	School support	0.797	0.636	1516.466	0.797	38.942 ***
Self-efficacy of pre-service teachers	School support	0.932	0.869	2891.191	0.343	16.873 ***
	Generative AI technology				0.636	31.299 ***
The content creation of pre-service teachers’ instructional design	School support	0.921	0.849	1622.973	0.062	2.468 *
	Generative AI technology				0.646	20.247 ***
	Self-efficacy of pre-service teachers				0.239	6.539 ***

Note: * *p* < 0.05, *** *p* < 0.001.

**Table 6 behavsci-15-00568-t006:** Analysis of the mediating effect between the generative AI technology and pre-service teacher self-efficacy.

Route	Intermediary Effect Value	Bootstrap SE	Boot CI Lower	Boot CI Upper	Proportion of Effect
Indirect Effect 1: School support → Generative AI technology → The content creation of pre-service teachers’ instructional design	0.5048	0.0362	0.4336	0.5762	66.00%
Indirect Effect 2: School support → Self-efficacy of pre-service teachers → The content creation of pre-service teachers’ instructional design	0.0803	0.0161	0.0499	0.1117	10.50%
Indirect Effect 3: School support → Generative AI technology → Self-efficacy of pre-service teachers → The content creation of pre-service teachers’ instructional design	0.1187	0.0260	0.0694	0.1729	15.52%
Total indirect effects	0.7038	0.0337	0.6355	0.7678	92.02%

## Data Availability

The datasets generated and analyzed during the current study are available from the corresponding author upon reasonable request.

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
