# Peer review of "How School Support Influences the Content Creation of Pre-Service Teachers’ Instructional Design"

_behavsci, 2025, doi:10.3390/bs15050568_

Round 1

Reviewer 1 Report

Comments and Suggestions for Authors

Title. The title expressed in the content of the manuscript, however it would be good if the authors could shorten it.

Abstract: I suggest to introduce two things: to start the abstract with a sentence that shows the relevance of the study; and despite saying that a survey was carried out, clarify what was the nature of the research methodology adopted, as well as identify what methods/instruments were used to collect data. You can rewrite the abstract, because it has not reached the word limit.

Introduction: The introduction presents a brief framework for the pertinence of the study, clarifying the authors' understanding about the instructional design, and ends with the three questions, well defined and clear, that guided the research.

Literature review: The theoretical framework focuses on the essential aspects that support the study distributed across three sections, both empirical and theoretical, based on diverse and actual references, ending with the proposal of a model for understanding the relationship between school support and teaching design content creation of pre-service teachers. From this review, the three hypotheses of the study(H1, H2, H3) stand out.

Methods: This item is well written, clear and justified. I would like to highlight the reference to the attention taken with the final questionnaire, using of a pre-questionnaire and the impact of its use on the final questionnaire, and in identifying quality criteria in particular to reliability and validity; the reference to the way in which the four group variables were measured, justified with support on literature; and the description of the data analysis, although short, sufficient.

Results, Discussion & Conclusions: The results follow the requirements demanded by a quantitative study, using descriptive statistics, with an adequate and complete interpretation of these results, grounded in literature. The discussion of the results is very interesting and very important for pre-service teacher education.

Despite the discussion carried out around the three hypotheses stated, and since three research questions were raised at the beginning of the manuscript (p.61-54), I propose that to answer to these questions should be highlighted in some way. Possibly in the final part of the conclusions.

This study pointed out some interesting findings, that I suggest, in future, to follow a qualitative approach to a focus group using interviews that makes it possible to understand some of the results obtained. Many congratulations to the authors.

Author Response

Comments 1

-Title. The title expressed in the content of the manuscript, however it would be good if the authors could shorten it.

Response 1

Thank you for the pertinent suggestions. We agree with your point of view. The title of this study is revised to:

 “How school support influence the content creation of pre-service teachers' instructional design.”

Comments 2

-Abstract: I suggest to introduce two things: to start the abstract with a sentence that shows the relevance of the study; and despite saying that a survey was carried out, clarify what was the nature of the research methodology adopted, as well as identify what methods/instruments were used to collect data. You can rewrite the abstract, because it has not reached the word limit.

Response 2

Thank you very much for your attention to our manuscript and your valuable suggestions for revision. We strongly agree and plan to revise the summary based on these recommendations.

For the first suggestion, a sentence has been added at the beginning of the abstract to show the relevance of the study. The additions are as follows:

“In the rapidly evolving educational landscape, understanding how school support influences the content creation of pre-service teachers' instructional design is crucial for fostering effective teaching practices and sustainable professional development.” .Please refer to “Abstract”of manuscript.

As for the second suggestion, the research method: online questionnaire survey was adopted to conduct research and collect data. Therefore, in the “Abstract” content, this part is modified as follows:

“A total of 871 Chinese pre-service teachers were surveyed using an online questionnaire to assess school support, generative AI technology, self-efficacy, and instructional design content creation”.Please refer to “Abstract” of manuscript.

Comments 3

-Introduction: The introduction presents a brief framework for the pertinence of the study, clarifying the authors' understanding about the instructional design, and ends with the three questions, well defined and clear, that guided the research.

Response 3

Thank you very much for your valuable comments and careful review of our research work. With regard to your evaluation of the introduction,especially in the elaboration of the framework of research relevance and the understanding of instructional design.We thought about it further, then revised it in the introduction and asked four questions to ensure that each link was closely aligned with the research objective (as indicated in the revised manuscript):

“1.What impact does school support have on the content creation of pre-service teachers' instructional design?

...

4.Do the generative AI technology and pre-service teachers' self-efficacy play a chain mediating role in the influence of school support on the content creation of pre-service teachers' instructional design?”

Please refer to the “1.Introduction” of manuscript for the above changes.

Comments 4

-Literature review: The theoretical framework focuses on the essential aspects that support the study distributed across three sections, both empirical and theoretical, based on diverse and actual references, ending with the proposal of a model for understanding the relationship between school support and teaching design content creation of pre-service teachers. From this review, the three hypotheses of the study(H1, H2, H3) stand out.

Response 4

Thank you for your careful review of our manuscript and your valuable comments.For better reading and understanding, the literature review has been changed to literature review and research hypothesis, At the same time, we have included a short summary at the end of each relevant chapter, directly related to the corresponding hypothesis, so that readers can better understand how each section supports our research hypothesis, so relevant contents have been adjusted (as shown in the revised manuscript) :

Please refer to the 2.Literature review and 3.The Research hypothesesof manuscript for the above changes.

Comments 5

-Methods: This item is well written, clear and justified. I would like to highlight the reference to the attention taken with the final questionnaire, using of a pre-questionnaire and the impact of its use on the final questionnaire, and in identifying quality criteria in particular to reliability and validity; the reference to the way in which the four group variables were measured, justified with support on literature; and the description of the data analysis, although short, sufficient.

Response 5

Thank you very much for your valuable comments and careful review of our research work. We have carefully revised this part of the method to ensure the reliability and validity of the study (as shown in the revised draft manuscript) :

“This study consists of two stages: pre-investigation and formal investigation. (1) Pre-investigation stage. Questionnaires were distributed online to pre-service teachers of relevant disciplines in Zhaoqing College. A total of 88 questionnaires were recovered, of which 78 were effective, with an effective rate of 88.6. SPSS 26.0 software was used to analyze the reliability of the pre-survey data. The Klonbach αcoefficients of the overall scale and school support, the application of generative AI technology, the self-efficacy of pre-service teachers, and the content creation of pre-service teaching design were 0.923, 0.945, 0.803, 0.911 and 0.818, respectively. Among them, the factor load of individual items in the generative AI technology application table is<0.3, while the factor load of individual items in pre-service teachers' instructional design content creation is < 0.3. .....All the questionnaires were selected with the same option or regular), and 871 valid questionnaires were obtained, with an effective rate of 82.87%.”

Please refer to the “4.Methodology 4.1 Participants and Procedure” of manuscript for the above changes.

Comments 6

-Results, Discussion & Conclusions: The results follow the requirements demanded by a quantitative study, using descriptive statistics, with an adequate and complete interpretation of these results, grounded in literature. The discussion of the results is very interesting and very important for pre-service teacher education.

Response 6

Thank you very much for your careful review and high evaluation of the results, discussions and conclusions of our manuscript. We are encouraged to learn that you feel that this section meets the requirements of a quantitative study, uses descriptive statistics, provides a full and complete explanation of the results, and has a reasonable discussion based on the literature.Thank you in particular for pointing out that the discussion section is not only very interesting, but also important for pre-service teacher education.

As for the three parts of “Results, Discussion & Conclusions, we have also made relevant revisions so that readers can have a clearer understanding of our research (as shown in the revised draft manuscript).

Please refer to the “5.Results, 6.Discussion and 9.Conclusions” of manuscript for the above changes.

Comments 7

-Despite the discussion carried out around the three hypotheses stated, and since three research questions were raised at the beginning of the manuscript (p.61-54), I propose that to answer to these questions should be highlighted in some way. Possibly in the final part of the conclusions.

Response 7

Thank you very much for your careful review of our manuscript and your valuable comments. We strongly agree.Therefore, we have carefully revised this part, and the part of “Conclusion” is revised as follows (as shown in the revised manuscript) :

“The following conclusions were drawn under the conditions of this study: this study proposes a chain mediation model to explore school support and the content creation of pre-service teachers' instructional design and its mechanism of action. The empirical research shows that, first, school support is significantly positively correlated with the content creation of pre-service teachers' instructional design; that is, the higher the level of school support is, the easier it is for pre-service teachers to achieve the content creation of instructional design. Second.....furthermore, school support promotes the application of generative AI technology, thereby impacting the level of self-efficacy of pre-service teachers and ultimately affecting the content creation of pre-service teachers' instructional design.”

Please refer to the “9.Conclusions” of manuscript for the above changes.

Comments 8

-This study pointed out some interesting findings, that I suggest, in future, to follow a qualitative approach to a focus group using interviews that makes it possible to understand some of the results obtained. Many congratulations to the authors.

Response 8

Thank you for your affirmation and valuable advice. In response to your suggestion to further understand the results of this study through qualitative methods, particularly focus group interviews, we strongly agree that this approach can provide more insight into existing research.We plan to adopt this recommendation in future studies.

Reviewer 2 Report

Comments and Suggestions for Authors

The research carried out is of great interest to the sector. However there are some key challenges in the way this paper is written, key ones being:

  • The assumptions made as to the potential untested benefits of AI
  • Limited exploration of the nature of school support which is being considered here as a point of comparison
  • No link to the level of experience, the school culture and working environment of the preservice teacher have and the impact of that on self-efficacy

The conclusion does not clearly articulate the key findings. There is work to be done to ensure that the paper positions the research and pulls on the strong themes and their possible applications.

Comments on the Quality of English Language

No comments

Author Response

Comments 1

-The English could be improved to more clearly express the research.

Response 1

We are grateful for the reviewer's comment regarding the English expression in our manuscript. We recognize that clear communication is crucial for ensuring that our research is accessible and understandable to the readers. In response to this feedback, we have meticulously reviewed and revised the language throughout the manuscript. We have paid particular attention to ensuring that our arguments are presented in a more concise and straightforward manner, while maintaining the technical accuracy of our descriptions. Additionally, we have utilized specialized academic writing tools to check for grammatical accuracy and stylistic consistency.

For example

(1)The Original Text of the “Abstract” in the manuscript : "The purpose of this study is to explore the influence path and mechanism of school support on the content creation of pre-service teachers' instructional design."

Revised Text: "This study aims to explore the influence pathways and mechanisms through which school support affects the content creation of pre-service teachers' instructional design."

(2)The Original Text of the 1.Introduction in the manuscript:“At present, generative artificial intelligence(AI) represented by ChatGPT, Sora, Wenxin Yiyi, and Xingfei Spark, based on large language models, has a profound impact on human knowledge production, information acquisition, education mode, etc., and has gradually evolved into an application revolution in artificial intelligence education (Yang, Yang, & Tong, 2023). ...With the deepening integration of intelligent technology and education, the role of teachers needs to change, and pre-service teachers need to focus more on cultivating students' essential character, high-level thinking and complex problem solving ability, and become life mentors for students' growth.”

Revised Text: “Currently, generative artificial intelligence (AI) technologies, exemplified by ChatGPT, Sora, Wenxin Yiyi, and Xingfei Spark, which are based on large language models, have exerted a profound influence on human knowledge production, information acquisition, and educational paradigms. These technologies have gradually catalyzed an application revolution in the field of AI education (Yang, Yang, & Tong, 2023). ...As the integration of intelligent technology and education deepens, the role of teachers must evolve. Pre-service teachers, in particular, need to shift their focus towards fostering students' essential character traits, higher-order thinking skills, and complex problem-solving abilities. In this context, pre-service teachers are expected to become life mentors who guide students' holistic development.”

(3)The Original Text of the 7.Practical Implications” in the manuscript : "Current research expands the understanding of the multiple aspects of content creation in pre-service teachers' instructional design by identifying school support, application of generative AI technology, and pre-service teachers' self-efficacy as key determinants. ...which can not only improve teachers' instructional design ability, but also ensure that they better use technological tools in teaching, improve teaching quality and students' learning results. "

Revised Text: "The present study extends the understanding of the multifaceted nature of content creation in pre-service teachers' instructional design by identifying school support, the application of generative AI technology, and self-efficacy of pre-service teachers as key determinants....This not only improves their instructional design capabilities but also ensures more proficient use of technological tools in teaching, thereby enhancing teaching quality and student learning outcomes."

Comments 2

-The assumptions made as to the potential untested benefits of AI

Response 2

Thank you very much for your careful review of our manuscript and your valuable comments. We deeply share your concern regarding the hypothetical portion of the potential untested benefits of artificial intelligence (AI) and believe this is an important point that deserves further clarification and in-depth exploration.

We understand that untested hypotheses may raise questions about the reliability and practical application of the study's conclusions. Therefore, we will revisit this section in light of your suggestions to ensure that the discussion clearly distinguishes between proven advantages and potential possibilities that have not been adequately tested.

Therefore, this study complements the following work:

15 randomly selected participants (covering different genders, grades, and majors) from the original sample (N=871) conducted semi-structured interviews (30-45 minutes/person) focusing on the unexpected gains in the use of generative AI technology. The interview contents are from the aspects of collaborative innovation, education equity promotion, personalized learning support and so on.

  • On generative AI technology to promote collaborative innovation:
  1. "Can you describe a time when you worked with other normal students or teachers to design a lesson plan using generative AI technology tools? How is it different from traditional collaboration?"
  2. "How has generative AI technology changed your idea generation or decision-making process in instructional design?"

(2) Issues on generative AI technology to promote educational equity:

  1. In what ways, according to your observations, might generative AI technology bring about pedagogical improvements that have not yet been fully explored?
  2. "How do generative AI technology tools affect your focus on instructional design?"

(3) Personalized learning support under generative AI technology

  1. "Have generative AI technology tools ever analyzed your design history data to provide personalized feedback? How has this affected your teaching growth?"
  2. "How do you use the learner portraits generated by generative AI technology to adjust instructional design strategies?"

Through interviews, we uncovered several important insights about the potential untested benefits of generative AI technology tools. For example, In terms of personalized learning support, generative AI technology tools provide targeted instructional improvement recommendations by analyzing historical data.

These interview results not only validate our hypothesis, but also reveal more directions about the potential benefits of AI. We will further highlight these findings in the paper and reflect on their limitations.

Comments 3

-Limited exploration of the nature of school support which is being considered here as a point of comparison

Response 3

Thank you very much for your careful review of our study and your valuable comments. With regard to the insufficient exploration of the nature of school support in the research you pointed out, we fully agree that this is an important area that needs to be further strengthened.

To better respond to this comment, we plan to take the following steps to deepen our understanding and analysis of the nature of school support:

  1. Literature review: We will conduct a more detailed literature review, and the Ministry of Education will issue documents related to school support, such as the Action Plan for Teacher Education Revitalization (2018-2022), China's Education Modernization 2035, and the Education Informatization 2.0 Action Plan. These documents often cover teacher training, technical support, resource allocation, etc., and are well suited to underpin the different dimensions of school support.

The revised are as follows in the “2.Literature review of manuscript :

“The support of schools is crucial to improving the teaching design ability of pre-service teachers. For example, the document "China's Education Modernization 2035"(Ministry of Education of the People's Republic of China,2019)issued by the Ministry of Education emphasizes that systematic support measures, such as resource supply, technical support and professional development, should be adopted to encourage teachers to use modern information technology to improve teaching methods, so as to effectively promote the growth of pre-service teachers in teaching design. The "Action Plan for Artificial Intelligence Innovation in Colleges and Universities" (Ministry of Education of the People's Republic of China,2018)document requires schools to be equipped with advanced hardware facilities, but also need to develop suitable teaching software and platforms. Through such support, pre-service teachers are able to explore in practice how technology can be integrated into curriculum design.”

Please refer to the “2.Literature review 2.1 The relationship between school support and content creation of pre-service teachers' instructional design” of manuscript for the above changes

  1. Research methods:In the “Methodology” part and scale question design, ensure consistency with the guidelines in the policy document, so as to enhance the content validity. For example, when designing a questionnaire entry on school support, refer to specific measures in the policy, such as "The school and teachers will be helpful in the use of large model technology tools" or "My teachers will be very supportive of my work using large model technology in the creation of instructional design content."
  2. In-depth interviews: To gain a more nuanced understanding of the specific practices of school support, in-depth interviews were incorporated into this study. These interviews provided rich and concrete examples that elucidated the multifaceted nature of school support and its contribution to the development of pre-service teachers' instructional design abilities.

Interview work: 15 participants (covering different genders, grades and majors) were randomly selected from the original sample (N=871) in terms of administrative support, professional development support, emotional support, etc.

1.Administrative aspects:

(1)How does the school administration support your instructional design work? (e.g. resource allocation, scheduling, etc.)

(2)Are there any policies or procedures in place to help you better design your instruction?

2.Professional development support

(1)Does the school offer any training or seminars to improve your instructional design skills?

(2)How is the application of generative AI technology promoted and supported within the school?

3.Emotional support

(1) What kind of emotional support did you receive from the school when faced with challenges?

(2) How does the school atmosphere affect your confidence and innovation in instructional design?

Through interviews, school support is able to improve the teaching design level of pre-service teachers

Comments 4

-No link to the level of experience, the school culture and working environment of the preservice teacher have and the impact of that on self-efficacy

Response 4

Thank you very much for your careful review of this study and your valuable comments. With regard to the failure to adequately link pre-service teachers' experience level, school culture and work environment and their impact on self-efficacy, we fully agree that this is an important direction for improvement.

Studies have shown that teachers' experience level is closely related to self-efficacy (Tschannen-Moran & Hoy,2021). Tschannen-Moran, M., & Hoy, A. W. (2001). Teacher efficacy:  Capturing an elusive construct. Teaching and teacher education, 17(7), 783-805. This document can provide support for the argument.

 However, all of the participating pre-service teachers in our study were school students, which means that they had not yet accumulated much practical teaching experience. Therefore, we designed the study to focus more on their performance in the simulated teaching environment and their expectations for the future, rather than relying on their past teaching experience. Nevertheless, we believe that the influence of school culture and work environment on their self-efficacy is significant, especially in shaping future career attitudes.

Therefore, We have made the following modifications and work:

  1. literature review(as indicated in the revised manuscript):

“Ashton and Webb (1986) showed that Schools with supportive cultures that encourage collaboration and professional development tend to foster higher levels of self-efficacy among teachers, andThere were studies have shown that positive work environments characterized by strong leadership and adequate resources contribute to higher self-efficacy among educators(Klassen & Chiu, 2010; Ross & Bruce, 2007).”

Please refer to the “2.Literature review 2.3.The mediating role of pre-service teachers' self-efficacy” of manuscript for the above changes

  1. In-depth interview: Recently, this study increased the work of in-depth interview, designed and implemented a series of new in-depth interview questions, specifically to explore how these factors affect the self-efficacy of pre-service teachers. For example:

(1) How has your teaching experience and previous training affected your self-efficacy?

(2) How did the school culture and work environment contribute to or hinder your professional growth?

(3)What specific school support measures have enhanced your sense of self-efficacy in your teaching practice?

Through interviews, school-related support can indeed enhance the self-efficacy of pre-service teachers.

Comments 5

-The conclusion does not clearly articulate the key findings. There is work to be done to ensure that the paper positions the research and pulls on the strong themes and their possible applications.

Response 5

Thank you very much for your careful review of our paper and your valuable comments. We agree that the conclusions you point out do not clearly address key findings and do not adequately emphasize the research topic and its potential applications.

In order to better respond to your comments, we plan to make the following revisions to the results and discussion section (as indicated in the revised manuscript ) :

1.Modifications in the 5.Results section:

5.4 Direct Effect Test From Table 5, with school support as the independent variable and pre-service teachers' instructional design content creation as the dependent variable,... H1 is thus supported.”

5.5 Test of the Mediating Effects In this study, according to the results of correlation analysis, there was a significant correlation between school support, generative AI technology, pre-service teachers' self-efficacy and instructional design content creation, which supports further analysis.... the relationship between school support and pre-service teachers' instructional design (see Table 6).”

Please refer to the “5.Results 5.4 Direct Effect Test 5.5 Test of the Mediating Effects” of manuscript for the above changes.

2.The revised contents of the 9.Conclusion” part:

“The following conclusions were drawn under the conditions of this study: this study proposes a chain mediation model to explore school support and the content creation of pre-service teachers' instructional design and its mechanism of action....thereby impacting the level of self-efficacy of pre-service teachers and ultimately affecting the content creation of pre-service teachers' instructional design.”

Please refer to the “9.Conclusion” of manuscript for the above changes.

Reviewer 3 Report

Comments and Suggestions for Authors

The article addresses a relevant topic; therefore, it deserves our attention. There are, however, some improvements that could be made, especially in the research methodology section, which lacks some detail.

The literature review section could be divided into two sections: one in which the literature review is presented and another in which the model is presented.

In the literature review, it made sense to include a section on large model technology; discuss using Generative AI technology instead of large model technology.

The model could be improved, including information such as: 1. The numbered hypotheses (H1, H2, etc.) 2. The variables (independent, dependent, moderators) 3. Arrows indicating relationships between variables, with labels (such as “+” or “–” or the hypothesis numbers themselves).

Method section:

Maybe use Methodology as the section title instead of Method.

It would be important to include more information about the pre-test that was done and provide more details about the initial questionnaire and what was changed based on the pre-test.  It should also be said what the criteria were for eliminating 180 responses (line 189).

The discussion could be improved by integrating the literature with the findings to provide a coherent argument in favor of the hypothesis raised.

Appendix A should be referred to in the text to better understand its relevance.

Author Response

Comments 1

-The literature review section could be divided into two sections: one in which the literature review is presented and another in which the model is presented.

Response 1

Thank you very much for your careful review of our paper and your valuable comments. We will follow your suggestion to divide the literature review into two separate sections (see the revised manuscript ) :

Please refer to the “2.Literature review ” and “3.The research hypotheses”of manuscript for more detail.

Comments 2

-In the literature review, it made sense to include a section on large model technology; discuss using Generative AI technology instead of large model technology.

Response 2

Thank you very much for your careful review of our paper and your valuable comments. 

In the 1.Introduction section, we will discuss the chapter on “Generative AI technology”, replacing the previously mentioned “large model technology”. This section will explore how generative AI technology affects the instructional design process for pre-service teachers (see revised manuscript):

“Generative AI technology is a technology that uses artificial intelligence algorithms to automatically generate new data or content. ...researchers mostly rely on models such as Theory of Planned Behavior (TPB) and Technology Acceptance Model(TAM)(Ajzen, 1991ï¼›Davis, 1989; Pynoo et al., 2011). ”

Please refer to the “1.Introduction ” of manuscript for more detail.

Meanwhile, we also changed the relevant content of the paper from "large model technology" to "Generative AI technology"

Comments 3

-The model could be improved, including information such as: 1. The numbered hypotheses (H1, H2, etc.) 2. The variables (independent, dependent, moderators) 3. Arrows indicating relationships between variables, with labels (such as “+” or “–” or the hypothesis numbers themselves).

Response 3

Thank you very much for your careful review of our paper and your valuable comments. Based on your suggestions, we will make the following improvements to the research model:

1.Number hypotheses: Clearly list all hypotheses (e.g. H1, H2, etc.) and ensure that each hypothesis is closely related to the corresponding variables and relationships(See "3.The Research hypotheses" of the revised manuscript):

“3.The Research hypotheses In this study, school support may impact the content creation of pre-service teachers' instructional design. ...the following by hypotheses (see Figure 1) are proposed.”

Figure 1. Research hypothesis model.

Please refer to the “3.The Research hypotheses  of manuscript for more detail.

2.Description of variables: Detail independent variables (such as school support), dependent variables (such as instructional design content creation), and potentially moderating variables (such as self-efficacy). (see the final content of the "1.Introduction" and the modification of the "4.3.Data processing" the revised manuscript)

“The independent variable was school support, the dependent variable was pre-service teachers' teaching design content creation, and generative AI technology and pre-service teachers' self-efficacy played a chain mediating role. ”

Please refer to the “1.Introduction” of manuscript for the above changes.

“In this study, SPSS26.0 was used for quantitative analysis. ...and the dependent variable of the content creation of pre-service teachers' instructional design. ”

Please refer to the “4.3.Data processing of manuscript for the above changes.

3.Relationship diagram: Add arrows to represent relationships between variables, and mark these relationships with symbols (such as "+" or "-") or by directly labeling the hypothesis number. (See Figure 2 Content modification for details):

Figure 2. Chain mediation effect model.

Comments 4

-Method section:Maybe use Methodology as the section title instead of Method.

Response 4

Thank you very much for your careful review of our paper and your valuable comments. We agree with your suggestion to change the title of the section "Method" to "Methodology".

We believe this revision will help to more accurately reflect the depth and breadth of the section. In academic writing, "Methodology" is often used to describe the overall framework of research methodologies, the reasons for choosing those methodologies, and the specific steps for how to implement them. In contrast, "Method" may seem more specific and technical, focusing on the practical level.The modification is as follows:

Please refer to the“4.Methodologyof manuscript.

Comments 5

-It would be important to include more information about the pre-test that was done and provide more details about the initial questionnaire and what was changed based on the pre-test.  It should also be said what the criteria were for eliminating 180 responses (line 189).

Response 5

Thank you very much for your careful review of our paper and your valuable comments. We agree with your questions about the need to provide more information about the pretest, the details of the initial questionnaire and the changes made based on the pretest, and to explain the criteria for deleting 180 responses.

In order to better respond to your comments, we plan to revise the relevant sections as follows:

1.In the pre-test and formal test(see "4.1.Participants and Procedure" of the revised manuscript):

“This study consists of two stages: pre-investigation and formal investigation. (1) Pre-investigation stage. Questionnaires were distributed online to pre-service teachers of relevant disciplines in Zhaoqing College. A total of 88 questionnaires were recovered, of which 78 were effective, with an effective rate of 88.6. SPSS 26.0 software was used to analyze the reliability of the pre-survey data. The Klonbach α coefficients of the overall scale and school support, the application of generative AI technology, the self-efficacy of pre-service teachers, and the content creation of pre-service teaching design were 0.923, 0.945, 0.803, 0.911 and 0.818, respectively. Among them, the factor load of individual items in the generative AI technology application table is < 0.3, while the factor load of individual items in pre-service teachers' instructional design content creation is < 0.3. ... (2) Formal investigation stage. The questionnaires were distributed in two ways: On the one hand, the questionnaires were distributed online to the pre-service teachers of relevant disciplines in Zhaoqing College, and a total of 549 questionnaires were collected; On the other hand, a total of 502 questionnaires were collected through relevant teachers distributing questionnaires in class. ...and 871 valid questionnaires were obtained, with an effective rate of 82.87%.”

Please refer to the“4.1.Participants and Procedure”of manuscript for more detail.

2.Delete the criteria for 180 responses

In the data analysis section, we will clarify the specific criteria for deleting 180 responses. Exclusion criteria: (1) There are missing or empty questions in the questionnaire; (2) The same option is selected for the whole questionnaire or it is regular (see the revised manuscript) :

“Finally, 180 invalid questionnaires were eliminated (invalid questionnaires were eliminated based on missing and empty questions; All the questionnaires were selected with the same option or regular), and 871 valid questionnaires were obtained, with an effective rate of 82.87%.”

Please refer to the“4.1.Participants and Procedure”of manuscript.

Comments 6

-The discussion could be improved by integrating the literature with the findings to provide a coherent argument in favor of the hypothesis raised.

Response 6

Thank you for your valuable comments. In response to your suggestion that there needs to be a better combination of Literature review”and research findings in the Discussion section to support our hypothesis, we fully agree and plan to make the following improvements:

Please refer to the“2.Literature review”and “6.Discussion of manuscript.

Comments 7

-Appendix A should be referred to in the text to better understand its relevance.

Response 7

Thank you very much for your careful review of our paper and your valuable comments. We agree with your suggestion that Appendix A needs to be referenced in the text to help readers better understand its relevance.

In order to better respond to your comments, we plan to make the following revisions to the text:We will add specific references to Appendix A where appropriate and explain how they relate to the content of the text. For example:

Please refer to the“4.2.1 School support scale”of manuscript:The social impact scale compiled by Thompson, Higgins, and Howell(1991)and revised by Venkatesh, Morris, Davis, and Davis(2003)was adapted according to the use of generative AI technology by teachers and students in schools(see Appendix A).

Please refer to the“4.2.2 Pre-service teacher self-efficacy scale”of manuscript:Based on the situation of pre-service teachers using educational AIGC to create instructional design content, this study has adapted five items(see Appendix A).

Please refer to the“4.2.3 Scale of application of generative AI technology”of manuscript:The human-centered artificial intelligence application scale developed by Berretta, Tausch, Peifer, and Kluge(2023) and Jaboob, Hazaimeh, and Al-Ansi (2023) was adapted according to the application of generative AI technology in teaching design content creation. The revised scale contains four items(see Appendix A).

Please refer to the“4.2.4 Instructional design content creation scale”of manuscript:The instructional design content creation questionnaire was compiled by Jaboob, et al., (2023) , and a total of 4 items were set to design the questionnaire(see Appendix A).

Reviewer 4 Report

Comments and Suggestions for Authors

The paper was interesting to read and addressed a gap-in-knowledge. I would respectively suggest the following: Lines 37 – 41: Very insightful and highly relative statement Line 44: “create situation" direct quote? Line 45 – 47: The innovation degree of "creating context" in teaching design is an important standard to measure teachers' creativity (Lai et al., 2017). Also needs to include page number – direct quote. Lines 54-56: At present, in the research field of "educators' acceptance of large models", re-searchers mostly rely on models such as planned behavior theory and technology acceptance. direct quote? Lines 65-66: . . . this study takes the pre-service teachers of Zhaoqing University in Guangdong Province as the research object and examines the relationships among school support . . . Possibly a better word than takes and possibly provide the number of pre-service teachers involved in the research project. Lines 90 – 91: It is concluded that school support is a necessary factor to improve teachers' content creation level in instructional design. Re-phrase this statement. For example, The conclusion derived from this research project suggests that . . . Line 102-105: A large number of relevant studies have shown that organizational support is an important factor affecting teachers' work attitude (Forlin et al.,2014), among which school support is an integral part of organizational support, so school sup-104 port will affect teachers' behavior in applying large model technology. You refer to a large number of relevant studies but only reference one. Line 108: The research of Lu et al(2022)also confirmed the promoting relationship between – expression. Lines 108 – 118: The research of Lu et al(2022)also confirmed the promoting relationship between large model technology and application on content creation of instructional design. The research results show that other elements, such as large model technology and application, school organization arrangement and resource provision, can meet the development needs of future teachers in a targeted manner, provide a creative supporting environment, promote the generation of innovative activities, and ultimately help to improve the level of content creation of teaching design for future teachers (Wolff et al.,2015). Therefore, the application of large model technology has the potential to improve the level of teaching design content creation of pre-service teachers. Therefore, the research hypothesis H2b is proposed: the application of large model technology positively predicts the content creation of pre-service teachers' instructional design. This paragraph needs to be re-written to present a stronger link with the paragraph above and a review of the use of, therefore. Line 125 - Bandura et al (1977) check APA 7 referencing. Line 131 - Kelm et al (2012) check APA 7 referencing. Line 171 – possibly provide an overview of your 4 hypotheses before unpacking them to give the reader an understanding of your ideas. Line 198 – 199: Thompson et al(1991) and revised by Venkatesh et al(2003)check APA 7 referencing. Lines 201-211: Tschannen-Moran et al (2001) and the General Teacher Self-Efficacy Scale designed by Schwarzer et al (2008). check APA 7 reference. Lines 220 – 221: Berretta et 220 al (2023) and Jaboob et al (2023) check APA 7 referencing. Line 230 - Jaboob et al 230 (2023) check APA 7 referencing. Findings as opposed to Discussion Lines 361-385: Length of paragraph Lines 384 - These findings support the important role of school support in promoting 384 teachers' content creation in instructional design. – How? Paragraph lengths for Discussion need to be reviewed. Conclusion paragraph structure Line 584 - Informed Consent Statement: Informed consent was obtained from all subjects involved in the study. Do you have a copy of this form? School support (Thompson et al., 1991 and Venkatesh et al., 2003) – What is the purpose for these quotes. The survey forms could have been formatted a bit better to provide the reader with an understanding of the breakdown of the surveys. In their current format the forms are melded together.

Author Response

Comments 1

-Line 45 -47: The innovation degree of "creating context" in teaching design is an important standard to measure teachers' creativity (Lai et al., 2017). Also needs to include page number- direct quote.

Response 1

Thank you for your careful review of our paper and your valuable comments. We have carried out a careful check and taken the following measures according to the actual situation:

For line 44, 'create situation' : This phrase is not a direct quote, but our own expression, and we will rephrase it to avoid misunderstanding and ensure more clarity and accuracy (see revised manuscript) :

“In the teaching design, creating a relevant situation can link to expand students' thinking and improve students' ability to use knowledge.Creating a relevant situation is crucial for effective teaching.”

Comments 2

-Line 45 -47: The innovation degree of "creating context" in teaching design is an important standard to measure teachers' creativity (Lai et al., 2017). Also needs to include page number- direct quote.

Response 2

Thank you for your careful review of our paper and your valuable comments. We have checked carefully. It is confirmed that this paragraph is not a direct quote from Lai et al. (2017) or other literature, but rather a summative statement based on existing research and theoretical frameworks. To clarify this and avoid any misunderstanding, we will make appropriate adjustments to the paragraph to ensure clarity and originality (see the revised manuscript) :

“The innovation degree of the situation of creating in teaching design is an important standard to measure teachers' creativity (Lai & Chu, 2017).”

Comments 3

-Lines 54-56: At present, in the research field of "educators' acceptance of large models", re-searchers mostly rely on models such as planned behavior theory and technology acceptance. direct quote?

Response 3

Thank you for your careful review of our paper and your valuable comments. We have checked carefully. It is confirmed that the paragraph cites relevant literature. In order to follow your suggestions and ensure accuracy, we will make the following changes (see the revised manuscript):

“At present, in the research field of educators' acceptance of large models, researchers mostly rely on models such as Theory of Planned Behavior (TPB) and Technology Acceptance  Model (TAM) (Ajzen,1991; Davis,1989; Pynoo, et al,2011).”

Please refer to the “1.Introduction” of manuscript for the above changes and more detail.

The references are as follows:

Ajzen, I. (1991). The theory of planned behavior. Organizational Behavior and Human Decision Processes, 50(2), 179-211.

Davis, F. D. (1989). Perceived usefulness, perceived ease of use,  and user acceptance of information technology. MIS Quarterly, 13(3), 319-340.

Pynoo, B., Devolder, P., Tondeur, J., van Braak, J., Duyck, W., & Duyck,  P. (2011). Predicting secondary school teachers’ acceptance and use of a digital learning environment:  A cross-sectional study. Computers in Human Behavior, 27(1), 568-575.

Comments 4

-Lines 65-66: . . . this study takes the pre-service teachers of Zhaoqing University in Guangdong Province as the research object and examines the relationships among school support . . . Possibly a better word than takes and possibly provide the number of pre-service teachers involved in the research project.

Response 4

Thank you very much for your careful review of our paper and your valuable comments. We deeply agree and have made corresponding modifications.

In order to comply with your suggestions and ensure accurate representation, we will make the following changes (as shown in the revised manuscript) :

“Therefore, this study collected the feedback of 871 pre-service teachers from Zhaoqing University in Guangdong Province by means of online questionnaire. ...so as to provide certain references for the application and implementation of intelligent teaching technology for pre-service teachers in China.”

Please refer to the “1.Introduction” of manuscript for the above changes and more detail.

Comments 5

-Lines 90 – 91: It is concluded that school support is a necessary factor to improve teachers' content creation level in instructional design. Re-phrase this statement. For example, The conclusion derived from this research project suggests that . . .

Response 5

Thank you very much for your thorough review and valuable feedback on our manuscript.In response to your comments, we have made changes to the statement to improve its clarity and form. Following your example, this sentence is rephrased as follows:

“It is concluded that School support plays a crucial role in improving teachers' ability to create content in instructional design”

Please refer to the “2.1 The relationship between school support and content creation of pre-service teachers' instructional design” of manuscript for the above changes.

We feel that this revision is more in line with the academic tone of the paper and more accurately communicates our findings.

Comments 6

-Line 102-105: A large number of relevant studies have shown that organizational support is an important factor affecting teachers' work attitude (Forlin et al.,2014), among which school support is an integral part of organizational support, so school sup-104 port will affect teachers' behavior in applying large model technology. You refer to a large number of relevant studies but only reference one. Line 108: The research of Lu et al(2022)also confirmed the promoting relationship between – expression.

Response 6

Thank you very much for your careful review of our paper and your valuable comments. We deeply agree and have made corresponding modifications.According to your suggestion, we have added more relevant research literature to this paragraph to support our argument. The following is the added literature (as seen in the revised manuscript) :

“A large number of relevant studies have shown that organizational support is an important factor affecting teachers' work attitude (Ertmer,2005; Ross & Gray,2006; Hew & Brush, 2007; Forlin, Loreman, & Sharma,2014) , among which school support is an integral part of organizational support, so school support will affect teachers' behavior in applying generative AI technology. ”

Please refer to the “2.2 The mediating role of generative AI technology” of manuscript for the above changes.

The references are as follows:

Ross, J. A., & Gray, P. (2006). School leadership and student achievement:  The mediating effects of teacher beliefs. Canadian Journal of Education/Revue canadienne de l 'education, 798-822.

Ertmer, P. A. (2005). Teacher pedagogical beliefs: The final frontier in our quest for technology integration? . Educational technology research and development, 53(4), 25-39.

Hew, K. F., & Brush, T. (2007). Integrating technology into K-12 teaching and learning:  Current knowledge gaps and recommendations for future research. Educational technology research and development, 55, 223-252.

Comments 7

-Lines 108 – 118: The research of Lu et al(2022)also confirmed the promoting relationship between large model technology and application on content creation of instructional design. The research results show that other elements, such as large model technology and application, school organization arrangement and resource provision, can meet the development needs of future teachers in a targeted manner, provide a creative supporting environment, promote the generation of innovative activities, and ultimately help to improve the level of content creation of teaching design for future teachers (Wolff et al.,2015). Therefore, the application of large model technology has the potential to improve the level of teaching design content creation of pre-service teachers. Therefore, the research hypothesis H2b is proposed: the application of large model technology positively predicts the content creation of pre-service teachers' instructional design. This paragraph needs to be re-written to present a stronger link with the paragraph above and a review of the use of, therefore.

Response 7

Thank you very much for your careful review of our paper and your valuable comments. We deeply agree and have made corresponding modifications:

“The study of Lu (2022) further verified the role of generative AI technology and its application in promoting the content creation of instructional design. ...Therefore, the generative AI technology has the potential to improve the level of teaching design content creation of pre-service teachers.”

Please refer to the “2.2 The mediating role of generative AI technology” of manuscript for the above changes and more detail.

Comments 8

-Line 125 - Bandura et al (1977) check APA 7 referencing.

Response 8

Thank you very much for your careful review of our paper and your valuable comments. We deeply agree and have made corresponding modifications:

“According to psychologist Bandura and Adams (1977), self-efficacy refers to an individual's confidence in his or her ability to complete a certain task or a certain type of task. ”

Comments 9

-Line 131 - Kelm et al (2012) check APA 7 referencing.

Response 9

Thank you very much for your careful review of our paper and your valuable comments. We deeply agree and have made corresponding modifications:

“Kelm & McIntosh(2012)”

Comments 10

-Line 171 – possibly provide an overview of your 4 hypotheses before unpacking them to give the reader an understanding of your ideas.

Response 10

Thank you very much for your careful review of our paper and your valuable comments. We deeply agree and have made corresponding modifications.

Modification and adjustment:

“3.The Research hypotheses In this study, school support may impact the content creation of pre-service teachers' instructional design. ...could be contingent upon the level the generative AI technology and pre-service teachers' self-efficacy”

Figure 1. Research hypothesis model.

“Hypothesis 1(H1):School group support has a positive predictive effect on the content creation of pre-service teachers' instructional design;

...

Hypothesis 4(H4):The application of generative AI technology and Self-efficacy of pre-service teachers play a chain mediating role in the relationship between school group and the content creation of pre-service teachers' instructional design.”

Please refer to the “3.The Research hypotheses” of manuscript for the above changes and more detail.

Comments 11

-Line 198 – 199: Thompson et al(1991) and revised by Venkatesh et al(2003)check APA 7 referencing.

Response 11

Thank you for your careful review of our paper and your valuable comments. We have made the following modifications:

“The social impact scale compiled by Thompson, Higgins, and Howell(1991)and revised by Venkatesh, Morris, Davis, and Davis(2003)was adapted according to the use of generative AI technology by teachers and students in schools(see Appendix A). ”

Comments 12

-Lines 201-211: Tschannen-Moran et al (2001) and the General Teacher Self-Efficacy Scale designed by Schwarzer et al (2008). check APA 7 reference.

Response 12

Thank you for your careful review of our paper and your valuable comments. We have made the following modifications:

“For the measurement of pre-service teachers' self-efficacy, please refer to the Teacher Self-Efficacy Scale developed by Tschannen-Moran and Hoy (2001) and the General Teacher Self-Efficacy Scale designed by Schwarzer and Hallum (2008). ”

Comments 13

-Lines 220 – 221: Berretta et 220 al (2023) and Jaboob et al (2023) check APA 7 referencing.

Response 13

Thank you for your careful review of our paper and your valuable comments. We have made the following modifications:

“The human-centered artificial intelligence application scale developed by Berretta, Tausch, Peifer, and Kluge(2023) and Jaboob, Hazaimeh, and Al-Ansi (2023) was adapted according to the application of generative AI technology in teaching design content creation. ”

Comments 14

-Line 230 - Jaboob et al 230 (2023) check APA 7 referencing.

Response 14

Thank you for your careful review of our paper and your valuable comments. We have made the following modifications:

“The human-centered artificial intelligence application scale developed by Berretta, Tausch, Peifer, and Kluge(2023) and Jaboob, Hazaimeh, and Al-Ansi (2023) was adapted according to the application of generative AI technology in teaching design content creation. ”

Comments 15

- Findings as opposed to Discussion Lines 361-385: Length of paragraph Lines 384 - These findings support the important role of school support in promoting 384 teachers' content creation in instructional design. – How? Paragraph lengths for Discussion need to be reviewed.

Response 15

Thank you very much for your careful review of our paper and your valuable comments. As for the problems you pointed out, we have checked and modified them accordingly:

“6.Discussion On the basis of previous studies, this study proposed a model with school support as the predictor, generative AI technology and self-efficacy of pre-service teachers as the mediating variables, and the content creation of pre-service teachers' instructional design as the result variables, showing a chain mediation effect...6.4 The Chain-Mediating Role of generative AI technology and Self-efficacy of Pre-service Teachers According to the standardized effect size of indirect effect of path 3 "School support →Generative AI technology→Self-efficacy of pre-service teachers→The content creation of pre-service teachers' instructional design",...Therefore, schools should emphasize providing relevant support to pre-service teachers and encourage them to apply generative AI technology to improve their instructional design content creation throughout the teaching process.”

Please refer to the “6.Discussion” of manuscript for the above changes and more detail.

Comments 16

- Conclusion paragraph structure Line 584 - Informed Consent Statement: Informed consent was obtained from all subjects involved in the study.  Do you have a copy of this form?

Response 16

Thank you for your careful review of our paper and your valuable comments. As for the issue of Informed Consent Form you mentioned, we fully understand its importance and are willing to make clarifications and additions on this basis:

  1. Implementation of informed consent(See the end of the reply draft for details)

In this study, all participants in the online questionnaire were clearly informed of the purpose, content and scope of data use. At the beginning of the questionnaire, we set up the informed consent statement page, which includes:

Research purpose: The purpose and significance of the research are explained.

Voluntary participation: Clearly inform the subject that participation is completely voluntary.

Confidentiality Pledge: Guarantee that all data collected will be kept strictly confidential and used only for academic research purposes.

Consent confirmation: Subjects need to click the "agree" button to continue filling in the questionnaire, otherwise they cannot enter the follow-up content.

This method conforms to the ethical requirements of online questionnaire survey and ensures the subjects' right to know and privacy protection.

Please refer to the“The introduction of the questionnaire”of the reply draft for more detail.

  1. Copy of informed consent

As this study adopts the form of online questionnaire, the informed consent of the subjects is completed by electronic means, so the traditional paper informed consent is not used.

  1. Additional notes on ethical review

In addition, this study has been approved by the ethical Review Committee, and the relevant ethical norms are strictly observed.

Comments 17

-School support (Thompson et al., 1991 and Venkatesh et al., 2003) – What is the purpose for these quotes. The survey forms could have been formatted a bit better to provide the reader with an understanding of the breakdown of the surveys. In their current format the forms are melded together.

Response 17

Thanks for your valuable comments, we will make the following adjustments:

1.About the purpose of the quote:

Our understanding is wrong. The scale given in the Appendix does not need to indicate references, as this part of references should be clearly explained in the research design.

2.Questionnaire format improved:

As for the format of the questionnaire you mentioned, after checking the relevant requirements of the journal, most articles of the contributing journal did not operate in this way, and after team discussion, they did not operate in this way for the time being. If you still suggest such operation, we will do the same. For example:

Guo, J., Abu Talib, M., Guo, B., Ren, J., & Liu, J. (2025). The Mediating Role of Satisfaction with Life and Social Interaction Anxiety in the Relationship Between Loneliness and Regulatory Emotional Self-Efficacy. Behavioral Sciences, 15(3), 392.

Gu, Q., & Zhao, M. (2025). The Chain Mediating Effects of Parent–Child Conflict and Screen Time on the Relationship Between Parental Phubbing and Problem Behaviors in Preschoolers. Behavioral Sciences, 15(2), 203.

The above two journal articles did not operate in this way. If you still suggest such operation, we will do the same.

The introduction of the questionnaire:

An investigation on the application of Education AIGC to STEM pre-service teacher training

Dear students,

Hello! This is a questionnaire about the application of educational AIGC to pre-service teacher training, aiming to understand the current situation of people's use of AIGC. The filling process takes about 15-25 minutes, your careful response is very important to us, please choose the most suitable answer according to your actual situation.

The results of the survey are for academic research only and confidential, and will not have any negative impact on your study and life. Your participation in this survey is voluntary and you have the right to stop answering and withdraw from the survey at any time.

If you agree with the above instructions, please continue to answer the questionnaire; If you do not agree, please close this questionnaire page. Thank you again for your attention and support!

AIGC (AI Generated Content) : A technology that uses artificial intelligence technology to generate natural language text, images, audio, video and other content.

STEM pre-service teachers: normal university students who specialize in the teaching of Science, Technology, Engineering and Mathematics.

Round 2

Reviewer 2 Report

Comments and Suggestions for Authors

I accept the changes made by the author and I am happy to accept the paper in its new form.